# Learned Step Size Quantization

**Steven K. Esser** *, **Jeffrey L. McKinstry, Deepika Bablani,**
**Rathinakumar Appuswamy, Dharmendra S. Modha**

IBM Research
San Jose, California, USA

## Abstract

Deep networks run with low precision operations at inference time offer power and space advantages over high precision alternatives, but need to overcome the challenge of maintaining high accuracy as precision decreases. Here, we present a method for training such networks, Learned Step Size Quantization, that achieves the highest accuracy to date on the ImageNet dataset when using models, from a variety of architectures, with weights and activations quantized to 2-, 3- or 4-bits of precision, and that can train 3-bit models that reach full precision baseline accuracy. Our approach builds upon existing methods for learning weights in quantized networks by improving how the quantizer itself is configured. Specifically, we introduce a novel means to estimate and scale the task loss gradient at each weight and activation layer's quantizer step size, such that it can be learned in conjunction with other network parameters. This approach works using different levels of precision as needed for a given system and requires only a simple modification of existing training code.

## 1 Introduction

Deep networks are emerging as components of a number of revolutionary technologies, including image recognition (Krizhevsky et al., 2012), speech recognition (Hinton et al., 2012), and driving assistance (Xu et al., 2017). Unlocking the full promise of such applications requires a system perspective where task performance, throughput, energy-efficiency, and compactness are all critical considerations to be optimized through co-design of algorithms and deployment hardware. Current research seeks to develop methods for creating deep networks that maintain high accuracy while reducing the precision needed to represent their activations and weights, thereby reducing the computation and memory required for their implementation. The advantages of using such algorithms to create networks for low precision hardware has been demonstrated in several deployed systems (Esser et al., 2016; Jouppi et al., 2017; Qiu et al., 2016).

It has been shown that low precision networks can be trained with stochastic gradient descent by updating high precision weights that are quantized, along with activations, for the forward and backward pass (Courbariaux et al., 2015; Esser et al., 2016). This quantization is defined by a mapping of real numbers to the set of discrete values supported by a given low precision representation (often integers with 8-bits or less). We would like a mapping for each quantized layer that maximizes task performance, but it remains an open question how to optimally achieve this.

To date, most approaches for training low precision networks have employed uniform quantizers, which can be configured by a single step size parameter (the width of a quantization bin), though more complex nonuniform mappings have been considered (Polino et al., 2018). Early work with low precision deep networks used a simple fixed configuration for the quantizer (Hubara et al., 2016; Esser et al., 2016), while starting with Rastegari et al. (2016), later work focused on fitting the quantizer to the data, either based on statistics of the data distribution (Li & Liu, 2016; Zhou et al., 2016; Cai et al., 2017; McKinstry et al., 2018) or seeking to minimize quantization error during training (Choi et al., 2018c; Zhang et al., 2018). Most recently, work has focused on using backpropagation with

---

*corresponding author sesser@us.ibm.com

Table 1: Comparison of low precision networks on ImageNet. Techniques compared are QIL (Jung et al., 2018), FAQ (McKinstry et al., 2018), LQ-Nets (Zhang et al., 2018), PACT (Choi et al., 2018b), Regularization (Choi et al., 2018c), and NICE (Baskin et al., 2018).

| Network | Method | Top-1 Accuracy @ Precision | | | | Top-5 Accuracy @ Precision | | | |
|---|---|---|---|---|---|---|---|---|---|
| | | 2 | 3 | 4 | 8 | 2 | 3 | 4 | 8 |
| ResNet-18 | | *Full precision: 70.5* | | | | *Full precision: 89.6* | | | |
| | LSQ (Ours) | **67.6** | **70.2** | **71.1** | **71.1** | **87.6** | **89.4** | **90.0** | **90.1** |
| | QIL | 65.7 | 69.2 | 70.1 | | | | | |
| | FAQ | | | 69.8 | 70.0 | | | 89.1 | 89.3 |
| | LQ-Nets | 64.9 | 68.2 | 69.3 | | 85.9 | 87.9 | 88.8 | |
| | PACT | 64.4 | 68.1 | 69.2 | | 85.6 | 88.2 | 89.0 | |
| | NICE | | 67.7 | 69.8 | | | 87.9 | 89.21 | |
| | Regularization | 61.7 | | 67.3 | 68.1 | 84.4 | | 87.9 | 88.2 |
| ResNet-34 | | *Full precision: 74.1* | | | | *Full precision: 91.8* | | | |
| | LSQ (Ours) | **71.6** | **73.4** | **74.1** | **74.1** | **90.3** | **91.4** | **91.7** | **91.8** |
| | QIL | 70.6 | 73.1 | 73.7 | | | | | |
| | LQ-Nets | 69.8 | 71.9 | | | 89.1 | 90.2 | | |
| | NICE | | 71.7 | 73.5 | | | 90.8 | 91.4 | |
| | FAQ | | | 73.3 | 73.7 | | | 91.3 | 91.6 |
| ResNet-50 | | *Full precision: 76.9* | | | | *Full precision: 93.4* | | | |
| | LSQ (Ours) | **73.7** | **75.8** | **76.7** | **76.8** | **91.5** | **92.7** | 93.2 | **93.4** |
| | PACT | 72.2 | 75.3 | 76.5 | | 90.5 | 92.6 | 93.2 | |
| | NICE | | 75.1 | 76.5 | | | 92.3 | **93.3** | |
| | FAQ | | | 76.3 | 76.5 | | | 92.9 | 93.1 |
| | LQ-Nets | 71.5 | 74.2 | 75.1 | | 90.3 | 91.6 | 92.4 | |
| ResNet-101 | | *Full precision: 78.2* | | | | *Full precision: 94.1* | | | |
| | LSQ (Ours) | **76.1** | **77.5** | **78.3** | **78.1** | **92.8** | **93.6** | **94.0** | **94.0** |
| ResNet-152 | | *Full precision: 78.9* | | | | *Full precision: 94.3* | | | |
| | LSQ (Ours) | **76.9** | **78.2** | **78.5** | **78.5** | **93.2** | **93.9** | **94.1** | **94.2** |
| | FAQ | | | 78.4 | **78.5** | | | **94.1** | 94.1 |
| VGG-16bn | | *Full precision: 73.4* | | | | *Full precision: 91.5* | | | |
| | LSQ (Ours) | **71.4** | **73.4** | **74.0** | 73.5 | **90.4** | **91.5** | **92.0** | **91.6** |
| | FAQ | | | 73.9 | **73.7** | | | 91.7 | **91.6** |
| Squeeze Next-23-2x | LSQ (Ours) | **53.3** | **63.7** | **67.4** | **67.0** | **77.5** | **85.4** | **87.8** | **87.7** |

stochastic gradient descent to learn a quantizer that minimizes task loss (Zhu et al., 2016; Mishra & Marr, 2017; Choi et al., 2018b;a; Jung et al., 2018; Baskin et al., 2018; Polino et al., 2018).

While attractive for their simplicity, fixed mapping schemes based on user settings place no guarantees on optimizing network performance, and quantization error minimization schemes might perfectly minimize quantization error and yet still be non optimal if a different quantization mapping actually minimizes task error. Learning the quantization mapping by seeking to minimize task loss is appealing to us as it directly seeks to improve on the metric of interest. However, as the quantizer itself is discontinuous, such an approach requires approximating its gradient, which existing methods have done in a relatively coarse manner that ignore the impact of transitions between quantized states (Choi et al., 2018b;a; Jung et al., 2018).

Here, we introduce a new way to learn the quantization mapping for each layer in a deep network, *Learned Step Size Quantization* (LSQ), that improves on prior efforts with two key contributions. First, we provide a simple way to approximate the gradient to the quantizer step size that is sensitive to quantized state transitions, arguably providing for finer grained optimization when learning the step size as a model parameter. Second, we propose a simple heuristic to bring the magnitude of step size updates into better balance with weight updates, which we show improves convergence. The overall approach is usable for quantizing both activations and weights, and works with existing methods for backpropagation and stochastic gradient descent. Using LSQ to train several network architectures on

the ImageNet dataset, we demonstrate significantly better accuracy than prior quantization approaches (Table 1) and, for the first time that we are aware of, demonstrate the milestone of 3-bit quantized networks reaching full precision network accuracy (Table 4).

## 2 METHODS

We consider deep networks that operate at inference time using low precision integer operations for computations in convolution and fully connected layers, requiring quantization of the weights and activations these layers operate on. Given data to quantize $v$, quantizer step size $s$, the number of positive and negative quantization levels $Q_P$ and $Q_N$, respectively, we define a quantizer that computes $\bar{v}$, a quantized and integer scaled representation of the data, and $\hat{v}$, a quantized representation of the data at the same scale as $v$:

$$\bar{v} = \lfloor clip(v/s, -Q_N, Q_P) \rceil, \tag{1}$$

$$\hat{v} = \bar{v} \times s. \tag{2}$$

Here, $clip(z, r_1, r_2)$ returns $z$ with values below $r_1$ set to $r_1$ and values above $r_2$ set to $r_2$, and $\lfloor z \rceil$ rounds $z$ to the nearest integer. Given an encoding with $b$ bits, for unsigned data (activations) $Q_N = 0$ and $Q_P = 2^b - 1$ and for signed data (weights) $Q_N = 2^{b-1}$ and $Q_P = 2^{b-1} - 1$.

For inference, $\bar{w}$ and $\bar{x}$ values can be used as input to low precision integer matrix multiplication units underlying convolution or fully connected layers, and the output of such layers then rescaled by the step size using a relatively low cost high precision scalar-tensor multiplication, a step that can potentially be algebraically merged with other operations such as batch normalization (Figure 1).

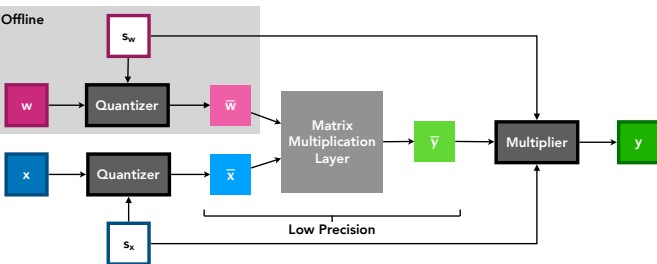

Figure 1: Computation of a low precision convolution or fully connected layer, as envisioned here.

### 2.1 STEP SIZE GRADIENT

LSQ provides a means to learn $s$ based on the training loss by introducing the following gradient through the quantizer to the step size parameter:

$$\frac{\partial \hat{v}}{\partial s} = \begin{cases} -v/s + \lfloor v/s \rceil & \text{if } -Q_N < v/s < Q_P \\ -Q_N & \text{if } v/s \leq -Q_N \\ Q_P & \text{if } v/s \geq Q_P \end{cases} \tag{3}$$

This gradient is derived by using the straight through estimator (Bengio et al., 2013) to approximate the gradient through the round function as a pass through operation (though leaving the round itself in place for the purposes of differentiating down stream operations), and differentiating all other operations in Equations 1 and 2 normally.

This gradient differs from related approximations (Figure 2), which instead either learn a transformation of the data that occurs completely prior to the discretization itself (Jung et al., 2018), or estimate the gradient by removing the round operation from the forward equation, algebraically canceling terms, and then differentiating such that $\partial \hat{v}/\partial s = 0$ where $-Q_N < v/s < Q_P$ (Choi et al., 2018b;a). In both such previous approaches, the relative proximity of $v$ to the transition point between quantized states does not impact the gradient to the quantization parameters. However, one can reason that the

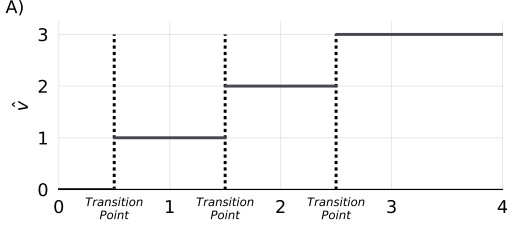 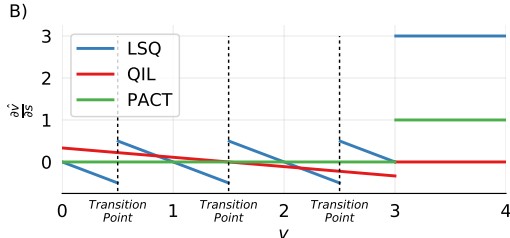

Figure 2: Given $s = 1$, $Q_N = 0$, $Q_P = 3$, A) quantizer output and B) gradients of the quantizer output with respect to step size, $s$, for LSQ, or a related parameter controlling the width of the quantized domain (equal to $s(Q_P + Q_N)$) for QIL (Jung et al., 2018) and PACT (Choi et al., 2018b). The gradient employed by LSQ is sensitive to the distance between $v$ and each transition point, whereas the gradient employed by QIL (Jung et al., 2018) is sensitive only to the distance from quantizer clip points, and the gradient employed by PACT (Choi et al., 2018b) is zero everywhere below the clip point. Here, we demonstrate that networks trained with the LSQ gradient reach higher accuracy than those trained with the QIL or PACT gradients in prior work.

closer a given $v$ is to a quantization transition point, the more likely it is to change its quantization bin ($\bar{v}$) as a result of a learned update to $s$ (since a smaller change in $s$ is required), thereby resulting in a large jump in $\hat{v}$. Thus, we would expect $\partial\hat{v}/\partial s$ to increase as the distance from $v$ to a transition point decreases, and indeed we observe this relationship in the LSQ gradient. It is appealing that this gradient naturally falls out of our simple quantizer formulation and use of the straight through estimator for the round function.

For this work, each layer of weights and each layer of activations has a distinct step size, represented as an fp32 value, initialized to $2\langle|v|\rangle/\sqrt{Q_P}$, computed on either the initial weights values or the first batch of activations, respectively.

## 2.2 STEP SIZE GRADIENT SCALE

It has been shown that good convergence is achieved during training where the ratio of average update magnitude to average parameter magnitude is approximately the same for all weight layers in a network (You et al., 2017). Once learning rate has been properly set, this helps to ensure that all updates are neither so large as to lead to repeated overshooting of local minima, nor so small as to lead to unnecessarily long convergence time. Extending this reasoning, we consider that each step size should also have its update magnitude to parameter magnitude proportioned similarly to that of weights. Thus, for a network trained on some loss function $L$, the ratio

$$R = \frac{\nabla_s L}{s} \bigg/ \frac{\|\nabla_w L\|}{\|w\|} \tag{4}$$

should on average be near 1, where $\|z\|$ denotes the $l_2$-norm of $z$. However, we expect the step size parameter to be smaller as precision increases (because the data is quantized more finely), and step size updates to be larger as the number of quantized items increases (because more items are summed across when computing its gradient). To correct for this, we multiply the step size loss by a gradient scale, $g$, where for weight step size $g = 1/\sqrt{N_W Q_P}$ and for activation step size $g = 1/\sqrt{N_F Q_P}$, where $N_W$ is the number of weights in a layer and $N_f$ is the number of features in a layer. In section 3.4 we demonstrate that this improves trained accuracy, and we provide reasoning behind the specific scales chosen in the Section A of the Appendix.

## 2.3 TRAINING

Model quantizers are trained with LSQ by making their step sizes learnable parameters with loss gradient computed using the quantizer gradient described above, while other model parameters can be trained using existing techniques. Here, we employ a common means of training quantized networks (Courbariaux et al., 2015), where full precision weights are stored and updated, quantized weights

and activations are used for forward and backward passes, the gradient through the quantizer round function is computed using the straight through estimator (Bengio et al., 2013) such that

$$\frac{\partial \hat{v}}{\partial v} = \begin{cases} 1 & \text{if } -Q_N < v/s < Q_P \\ 0 & \text{otherwise,} \end{cases} \tag{5}$$

and stochastic gradient descent is used to update parameters.

For simplicity during training, we use $\hat{v}$ as input to matrix multiplication layers, which is algebraically equivalent to the previously described inference operations. We set input activations and weights to either 2-, 3-, 4-, or 8-bit for all matrix multiplication layers except the first and last, which always use 8-bit, as making the first and last layers high precision has become standard practice for quantized networks and demonstrated to provide a large benefit to performance. All other parameters are represented using fp32. All quantized networks are initialized using weights from a trained full precision model with equivalent architecture before fine-tuning in the quantized space, which is known to improve performance (Sung et al., 2015; Zhou et al., 2016; Mishra & Marr, 2017; McKinstry et al., 2018).

Networks were trained with a momentum of 0.9, using a softmax cross entropy loss function, and cosine learning rate decay without restarts (Loshchilov & Hutter, 2016). Under the assumption that the optimal solution for 8-bit networks is close to the full precision solution (McKinstry et al., 2018), 8-bit networks were trained for 1 epoch while all other networks were trained for 90 epochs. The initial learning rate was set to 0.1 for full precision networks, 0.01 for 2-, 3-, and 4-bit networks and to 0.001 for 8-bit networks. All experiments were conducted on the ImageNet dataset (Russakovsky et al., 2015), using pre-activation ResNet (He et al., 2016), VGG (Simonyan & Zisserman, 2014) with batch norm, or SqueezeNext (Gholami et al., 2018). All full precision networks were trained from scratch, except for VGG-16bn, for which we used the pretrained version available in the PyTorch model zoo. Images were resized to $256 \times 256$, then a $224 \times 224$ crop was selected for training, with horizontal mirroring applied half the time. At test time, a $224 \times 224$ centered crop was chosen. We implemented and tested LSQ in PyTorch.

## 3 RESULTS

### 3.1 WEIGHT DECAY

We expect that reducing model precision will reduce a model's tendency to overfit, and thus also reduce the regularization in the form of weight decay necessary to achieve good performance. To investigate this, we performed a hyperparameter sweep on weight decay for ResNet-18 (Table 2), and indeed found that lower precision networks reached higher accuracy with less weight decay. Performance was improved by reducing weight decay by half for the 3-bit network, and reducing it by a quarter for the 2-bit network. We used these weight decay values for all further experiments.

Table 2: ResNet-18 top-1 accuracy for various weight decay values.

| Weight Decay | 2-bit | 3-bit | 4-bit | 8-bit |
|---:|:---:|:---:|:---:|:---:|
| $10^{-4}$ | 66.9 | 70.1 | **71.0** | **71.1** |
| $0.5 \times 10^{-4}$ | 67.3 | **70.2** | 70.9 | 71.1 |
| $0.25 \times 10^{-4}$ | **67.6** | 70.0 | 70.9 | 71.0 |
| $0.125 \times 10^{-4}$ | 67.4 | 66.9 | 70.8 | 71.0 |

### 3.2 COMPARISON WITH OTHER APPROACHES

We trained several networks using LSQ and compare accuracy with other quantized networks and full precision baselines (Table 1). To facilitate comparison, we only consider published models that quantize all convolution and fully connected layer weights and input activations to the specified precision, except for the first and last layers which may use higher precision (as for the LSQ models). In some cases, we report slightly higher accuracy on full precision networks than in their original publications, which we attribute to our use of cosine learning rate decay (Loshchilov & Hutter, 2016).

We found that LSQ achieved a higher top-1 accuracy than all previous reported approaches for 2-, 3-and 4- bit networks with the architectures considered here. For nearly all cases, LSQ also achieved the best-to-date top-5 accuracy on these networks, and best-to-date accuracy on 8-bit versions of these networks. In most cases, we found no accuracy advantage to increasing precision from 4-bit to 8-bit. It is worth noting that the next best low precision method (Jung et al., 2018) used progressive fine tuning (sequentially training a full precision to 5-bit model, then the 5-bit model to a 4-bit model, and so on), significantly increasing training time and complexity over our approach which fine tunes directly from a full precision model to the precision of interest.

It is interesting to note that when comparing a full precision to a 2-bit precision model, top-1 accuracy drops only 2.9 for ResNet-18, but 14.0 for SqueezeNext-23-2x. One interpretation of this is that the SqueezeNext architecture was designed to maximize performance using as few parameters as possible, which may have placed it at a design point extremely sensitive to reductions in precision.

## 3.3 ACCURACY VS. MODEL SIZE

For a model size limited application, it is important to choose the highest performing model that fits within available memory limitations. To facilitate this choice, we plot here network accuracy against corresponding model size (Figure 3).

We can consider the frontier of best performance for a given model size of the architectures considered here. On this metric, we can see that 2-bit ResNet-34 and ResNet-50 networks offer an absolute advantage over using a smaller network, but with higher precision. We can also note that at all precisions, VGG-16bn exists below this frontier, which is not surprising as this network was developed prior to a number of recent innovations in achieving higher performance with fewer parameters.

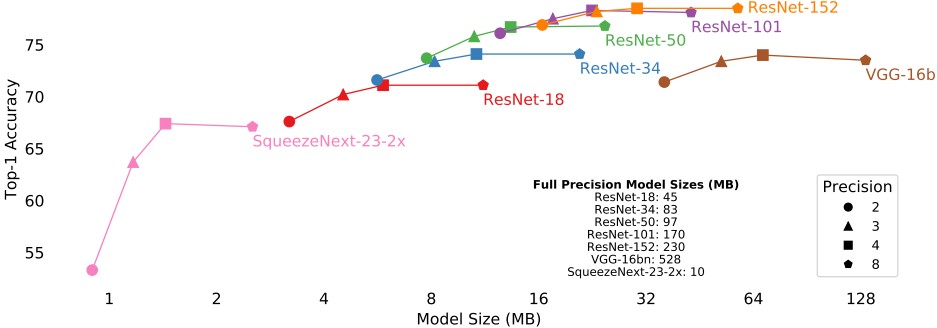

Figure 3: Accuracy vs. model size for the networks considered here show some 2-bit networks provide the highest accuracy at a given model size. Full precision model sizes are inset for reference.

## 3.4 STEP SIZE GRADIENT SCALE IMPACT

To demonstrate the impact of the step size gradient scale (Section 2.2), we measured $R$ (see Equation 4) averaged across $500$ iterations in the middle of the first training epoch for ResNet-18, using different step size gradient scales (the network itself was trained with the scaling as described in the methods to avoid convergence problems). With no scaling, we found that relative to parameter size, updates to step size were 2 to 3 orders of magnitude larger than updates to weights, and this imbalance increased with precision, with the 8-bit network showing almost an order of magnitude greater imbalance than the 2-bit network (Figure 4, left). Adjusting for the number of weights per layer ($g = 1/\sqrt{N_W}$), the imbalance between step size and weights largely went away, through the imbalance across precision remained (Figure 4, center). Adjusting for the number of number of weights per layer and precision ($g = 1/\sqrt{N_W Q_P}$), this precision dependent imbalance was largely removed as well (Figure 4, right).

We considered network accuracy after training a 2-bit ResNet-18 using different step size gradient scales (Table 3). Using the network with the full gradient scale ($g = 1/\sqrt{N_W Q_P}$ and $g = 1/\sqrt{N_F Q_P}$

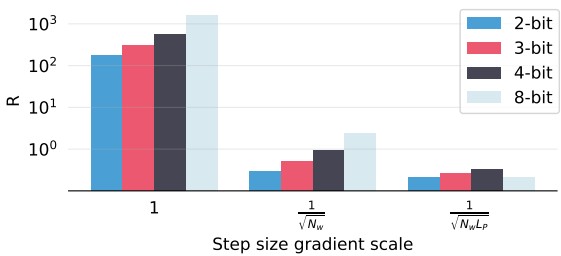

Figure 4: Relative parameter update magnitudes given different step size gradient scales. A gradient scale of $1/N_W Q_P$ better balances relative step size and weight gradient magnitudes (right vs. left).

for weight and activation step size respectively) as baseline, we found that adjusting only for weight and feature count led to a $0.3$ decrease in top-1 accuracy, and when no gradient scale was applied the network did not converge unless we dropped the initial learning rate. Dropping the initial learning rate in multiples of ten, the best top-1 accuracy we achieved using no gradient scale was $3.4$ below baseline, using an initial learning rate of $0.0001$. Finally, we found that using the full gradient scaling with an additional ten-fold increase or decrease also reduced top-1 accuracy. Overall, this suggests a benefit to our chosen heuristic for scaling the step size loss gradient.

Table 3: Top-1 accuracy for various gradient scale values and learning rates for 2-bit ResNet-18.

| Gradient scale | Learning Rate | Accuracy |
|---|---|---|
| $\mathbf{1/\sqrt{NQ_P}}$ | **0.01** | **67.6** |
| $1/\sqrt{N}$ | 0.01 | 67.3 |
| $1$ | 0.01 | Did not converge |
| $1$ | 0.0001 | 64.2 |
| $10/\sqrt{NQ_P}$ | 0.01 | 67.4 |
| $1/10\sqrt{NQ_P}$ | 0.01 | 67.3 |

### 3.5 COSINE LEARNING RATE DECAY IMPACT

We chose to use cosine learning rate decay in our experiments as it removes the need to select learning rate schedule hyperparameters, is available in most training frameworks, and does not increase training time. To facilitate comparison with results in other publications that use step-based learning rate decay, we trained a 2-bit ResNet-18 model with LSQ for 90 epochs, using an initial learning rate of $0.01$, which was multiplied by $0.1$ every 20 epochs. This model reached a top-1 accuracy of $67.2$, a reduction of $0.4$ from the equivalent model trained with cosine learning rate decay, but still marking an improvement of $1.5$ over the next best training method (see Table 1).

### 3.6 QUANTIZATION ERROR

We next sought to understand whether LSQ learns a solution that minimizes quantization error (the distance between $\hat{v}$ and $v$ on some metric), despite such an objective not being explicitly encouraged. For this purpose, for a given layer we define the final step size learned by LSQ as $\hat{s}$ and let $S$ be the set of discrete values $\{0.01\hat{s}, 0.02\hat{s}, ..., 20.00\hat{s}\}$. For each layer, on a single batch of test data we computed the value of $s \in S$ that minimizes mean absolute error, $\langle|(\hat{v}(s) - v)|\rangle$, mean square error, $\langle(\hat{v}(s) - v)^2\rangle$, and Kullback-Leibler divergence, $\int p(v) \log p(v) - \int p(v) \log q(\hat{v}(s))$ where $p$ and $q$ are probability distributions. For purposes of relative comparison, we ignore the first term of Kullback-Leibler divergence, as it does not depend on $\hat{v}$, and approximate the second term as $-\mathrm{E}[\log(q(\hat{v}(s)))]$, where the expectation is over the sample distribution.

For a 2-bit ResNet-18 model we found $\hat{s} = 0.949 \pm 0.206$ for activations and $\hat{s} = 0.025 \pm 0.019$ for weights (mean $\pm$ standard deviation). The percent absolute difference between $\hat{s}$ and the value of $s$ that minimizes quantization error, averaged across activation layers was $50\%$ for mean absolute error, $63\%$ for mean square error, and $64\%$ for Kullback-Leibler divergence, and averaged across weight layers, was $47\%$ for mean absolute error, $28\%$ for mean square error, and $46\%$ for Kullback-Leibler divergence. This indicates that LSQ learns a solution that does not in fact minimize quantization error. As LSQ achieves better accuracy than approaches that directly seek to minimize quantization error, this suggests that simply fitting a quantizer to its corresponding data distribution may not be optimal for task performance.

### 3.7 IMPROVEMENT WITH KNOWLEDGE-DISTILLATION

To better understand how well low precision networks can reproduce full precision accuracy, we combined LSQ with same-architecture knowledge distillation, which has been shown to improve low precision network training (Mishra & Marr, 2017). Specifically, we used the distillation loss function of Hinton et al. (2015) with temperature of 1 and equal weight given to the standard loss and the distillation loss (we found this gave comparable results to weighting the the distillation loss two times more or less than the standard loss on 2-bit ResNet-18). The teacher network was a trained full precision model with frozen weights and of the same architecture as the low precision network trained. As shown in Table 4, this improved performance, with top-1 accuracy increasing by up to 1.1 (3-bit ResNet-50), and with 3-bit networks reaching the score of the full precision baseline (see Table 1 for comparison). As a control, we also used this approach to distill from the full precision teacher to a full precision (initially untrained) student with the same architecture, which did not lead to an improvement in the student network accuracy beyond training the student alone. These results reinforce previous work showing that knowledge-distillation can help low precision networks catch up to full precision performance (Mishra & Marr, 2017).

Table 4: Accuracy for low precision networks trained with LSQ and knowledge distillation, which is improved over using LSQ alone, with 3-bit networks reaching the accuracy of full precision (32-bit) baselines (shown for comparison).

| Network | Top-1 Accuracy @ Precision | | | | | Top-5 Accuracy @ Precision | | | | |
|---|---|---|---|---|---|---|---|---|---|---|
| | 2 | 3 | 4 | 8 | *32* | 2 | 3 | 4 | 8 | *32* |
| ResNet-18 | 67.9 | 70.6 | 71.2 | 71.1 | *70.5* | 88.1 | 89.7 | 90.1 | 90.1 | *89.6* |
| ResNet-34 | 72.4 | 74.3 | 74.8 | 74.1 | *74.1* | 90.8 | 91.8 | 92.1 | 91.7 | *91.8* |
| ResNet-50 | 74.6 | 76.9 | 77.6 | 76.8 | *76.9* | 92.1 | 93.4 | 93.7 | 93.3 | *93.4* |

## 4 CONCLUSIONS

The results presented here demonstrate that on the ImageNet dataset across several network architectures, LSQ exceeds the performance of all prior approaches for creating quantized networks. We found best performance when rescaling the quantizer step size loss gradient based on layer size and precision. Interestingly, LSQ does not appear to minimize quantization error, whether measured using mean square error, mean absolute error, or Kullback-Leibler divergence. The approach itself is simple, requiring only a single additional parameter per weight or activation layer.

Although our goal is to train low precision networks to achieve accuracy equal to their full precision counterparts, it is not yet clear whether this goal is achievable for 2-bit networks, which here reached accuracy several percent below their full precision counterparts. However, we found that such 2-bit solutions for state-of-the-art networks are useful in that they can give the best accuracy for the given model size, for example, with an 8MB model size limit, a 2-bit ResNet-50 was better than a 4-bit ResNet-34 (Figure 3).

This work is a continuation of a trend towards steadily reducing the number of bits of precision necessary to achieve good performance across a range of network architectures on ImageNet. While it is unclear how far it can be taken, it is noteworthy that the trend towards higher performance at lower precision strengthens the analogy between artificial neural networks and biological neural networks,

which themselves employ synapses represented by perhaps a few bits of information (Bartol Jr et al., 2015) and single bit spikes that may be employed in small spatial and/or temporal ensembles to provide low bit width data representation. Analogies aside, reducing network precision while maintaining high accuracy is a promising means of reducing model size and increasing throughput to provide performance advantages in real world deployed deep networks.

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

## A   STEP SIZE GRADIENT SCALE DERIVATION

We compute our gradient scale value by first estimating $R$ (Equation 4), starting with the simple heuristic that for a layer with $N_W$ weights

$$\|w\|/s \approx \sqrt{N_W Q_P}. \tag{6}$$

To develop this approximation, we first note that the expected value of an $l_2$-norm should grow with the square root of the number of elements normalized. Next, we assume that where $Q_P = 1$, step size should be approximately equal to average weight magnitude so as to split the weight distribution into zero and non zero values in a roughly balanced fashion. Finally, we assume that for larger $Q_P$, step size should be roughly proportional to $\sqrt{1/Q_P}$, so that as the number of available quantized states increases, data between the clip points will be quantized more precisely, and the clip points themselves (equal to $sQ_N$ and $sQ_P$) will move further out to better encode outliers.

We also note that, in the expectation, $\|\nabla_w L\|$ and $\nabla_s L$ are of approximately the same order. This can be shown by starting from the chain rule

$$\nabla_s L = \sum_{i=1}^{N_W} \frac{\partial L}{\partial \hat{w}_i} \frac{\partial \hat{w}_i}{\partial s}, \tag{7}$$

then assuming $\partial \hat{w}_i / \partial s$ is reasonably close to 1 (see for example Figure 2), and treating all $\partial L / \partial \hat{w}_i$ as uncorrelated zero-centered random variables, to compute the following expectation across weights:

$$\mathrm{E}\left[\nabla_s L^2\right] \approx N_W \times \mathrm{E}\left[\frac{\partial L}{\partial \hat{w}}^2\right]. \tag{8}$$

By assuming $\partial \hat{w} / \partial w = 1$ for most weights, we similarly approximate

$$\mathrm{E}\left[\|\nabla_w L\|^2\right] \approx N_W \times \mathrm{E}\left[\frac{\partial L}{\partial \hat{w}}^2\right]. \tag{9}$$

Bringing all of this together, we can then estimate

$$R \approx \sqrt{N_W Q_P}. \tag{10}$$

Knowing this expected imbalance, we compute our gradient scale factor for weights by simply taking the inverse of $R$, so that $g$ is set to $1/\sqrt{N_W Q_P}$.

As most activation layers are preceded by batch normalization (Ioffe & Szegedy, 2015), and assuming updates to the learned batch normalization scaling parameter is the primary driver of changes to pre-quantization activations, we can use a similar approach to the above to show that there is an imbalance between step size updates and update driven changes to activations that grows with the number of features in a layer, $N_F$ as well as $Q_P$. Thus, for activation step size we set $g$ to $1/\sqrt{N_F Q_P}$.

## B   IMPLEMENTATION

In this section we provide pseudocode to facilitate the implementation of LSQ. We assume the use of automatic differentiation, as supported by a number of popular deep learning frameworks, where the desired operations for the training forward pass are coded, and the automatic differentiation engine computes the gradient through those operations in the backward pass.

Our approach requires two functions with non standard gradients, *gradscale* (Function 1) and *roundpass* (Function 2). We implement the custom gradients by assuming a function called *detach* that returns its input (unmodified) during the forward pass, and whose gradient during the backward pass is zero (thus detaching itself from the backward graph). This function is used in the form:

$$y = detach(x_1 - x_2) + x_2, \tag{11}$$

so that in the forward pass, $y = x_1$ (as the $x_2$ terms cancel out), while in the backward pass $\partial L/\partial x_1 = 0$ (as detach blocks gradient propagation to $x_1$) and $\partial L/\partial x_2 = \partial L/\partial y$. We also assume a function $nfeatures$ that given an activation tensor, returns the number of features in that tensor, and

$nweights$ that given a weight tensor, returns the number of weights in that tensor. Finally, the above are used to implement a function called *quantize*, which quantizes weights and activations prior to their use in each convolution or fully connected layer.

The pseudocode provided here is chosen for simplicity of implementation and broad applicability to many training frameworks, though more compute and memory efficient approaches are possible. This example code assumes activations are unsigned, but could be modified to quantize signed activations.

---

**Function 1** gradscale(x, scale):

```
# x: Input tensor
# scale: Scale gradient by this
yOut = x
yGrad = x × scale
y = detach(yOut - yGrad) + yGrad # Return yOut in forward, pass gradient to yGrad in backward
return y
```

---

**Function 2** roundpass(x):

```
# x: Input tensor
yOut = round(x) # Round to nearest
yGrad = x
y = detach(yOut - yGrad) + yGrad # Return yOut in forward, pass gradient to yGrad in backward
return y
```

---

**Function 3** quantize(v, s, p, isActivation):

```
# v: Input tensor
# s: Step size, a learnable parameter specific to weight or activation layer being quantized
# p: Quantization bits of precision
# isActivation: True if v is activation tensor,
#               False if v is weight tensor

# Compute configuration values
if isActivation:
    Qn = 0
    Qp = 2^p - 1
    gradScaleFactor = 1 / sqrt(nfeatures(v) × Qp)
else: # is weights
    Qn = -2^(p-1)
    Qp = 2^(p-1) - 1
    gradScaleFactor = 1 / sqrt(nweights(v) × Qp)

# Quantize
s = gradscale(s, gradScaleFactor)
v = v / s
v = clip(v, Qn, Qp)
vbar = roundpass(v)
vhat = vbar × s
return vhat
```

---

