# OpenReview forum: "LEARNED STEP SIZE QUANTIZATION"
_ICLR.cc/2020/Conference — Accept (Poster)_

### Official Review · AnonReviewer3 · 2019-10-22
**Official Blind Review #3**

**Rating:** 8

**Review:**

The paper is concerned with neural networks using low-precision operations. This is an important research are both for applications (e.g., deploying neural networks on embedded systems or other constrained hardware, or increasing throughput on a production system) and for theoretical reasons (e.g., potentially modelling the behavior of biological neural networks).

The paper makes two primary contributions: (1) It proposes a new approximation for the gradient of the quantized data with respect to the step size parameter (equation 3) based on the well-known straight through estimator. (2) It proposes a heuristic for scaling gradient updates based on the number of weights in a given layer and the quantization levels (clipping thresholds). Experimental results showing higher accuracy than previously reported approaches in a number of settings and for common metrics, and best-to-date results for other settings.

I recommend the paper for publication.

In terms of improving the paper further, the authors could expand their experiment section. E.g., how does their method compare to current SOTA knowledge-distillation methods; how does it deal with other architectures (recurrent models, attention, etc).

Minor notes on the manuscript:

In section 3.4, both the term “population size” as well as the symbol N_W come without any definition. While closer reading makes it clear what is meant is the number of weights per layer, a definition closer to their first usage would be good. This goes in particular for N_W which comes in several versions (N_W, N, nweights).

In section 3.7, the sentence starting with “We found that using this approach to distill” does not seem to fit within its context for this reader, as it describes “full-to-full” distillation. Perhaps it is meant to read “low precision student network”?

**Experience Assessment:**

I have published one or two papers in this area.

**Review Assessment: Checking Correctness Of Derivations And Theory:**

I assessed the sensibility of the derivations and theory.

**Review Assessment: Checking Correctness Of Experiments:**

I assessed the sensibility of the experiments.

**Review Assessment: Thoroughness In Paper Reading:**

I read the paper at least twice and used my best judgement in assessing the paper.

---

> ### Author Response · Authors · 2019-11-13
> **Response to review**
>
> We are very excited to apply this technique to more architectures and application domains.  Due to time constraints, we are unable to provide results for the current manuscript, but plan to tackle this area in future work.
>
> We wanted to keep our knowledge distillation approach simple to keep the primary focus of the paper on the quantizer step size training technique.  Thus, we have focused on same-architecture distillation (which has the advantage that it does not require training an additional network beyond what would be done for standard fine-tuning from a pre-trained initialization).  The only work we are aware of that takes a similar approach is Mishra and Marr, "Apprentice: Using knowledge distillation techniques to improve low-precision network accuracy" (who provided our inspiration to try this knowledge distillation variation), who report results on different precision levels than we report on here, making a direct comparison difficult (though notably, we show higher accuracy at lower precision when comparing with the closest comparable precision levels from Mishra and Marr).
>
> N_W and N_F are now defined where they first appear in section 2.2, reference to N has been replaced with reference to N_W and N_F (and we verified that "nweights", which refers to a function and is therefore subtly different from N_W, is defined where first used).
>
> We have clarified our use of full precision to full precision distillation, which is intended as a simple control experiment:
> "As a control, we also used this approach to distill from the full precision teacher to a full precision (initially untrained) student with the same architecture, which did not lead to an improvement in the student network accuracy beyond training the student alone."

---

### Official Review · AnonReviewer1 · 2019-10-26
**Official Blind Review #1**

**Rating:** 6

**Review:**


This paper trains low-precision network with quantized weights and quantized activation. The main idea is to split the scale and quantized values. Both scales and weights are updated with backprop and SGD. The paper presents excellent experimental results on ImageNet.

The paper is generally well written and easy to follow. However, there does exist quite some grammar errors, especially in abstract, which could be improved.

Moreover, I would like the authors to clarify some technical details. Are the scales s, so called step size in the paper, for every weight, every convolutional kernel, or very layer? How do you deal with BatchNorm?

What is the main benefits of the proposed quantization method in general? Is it for fast inference, fast training, or just memory compression? Do the authors see the real benefits in practice besides claiming the accuracy does not drop?

I would suggest the authors discuss and compare with XNOR network in detail. The proposed method looks similar.

I am wondering how the baseline methods are tuned. There are quite a few “tricks” like learning rate scheduler and weight decay, which I do consider them as contributions of the paper. But would baseline methods also benefit from more hyper-parameter tuning?

Minor issue, I donot get the explanation of eq (4), and it looks rather unnecessary. It sounds to me starting from a trained network and then train 90 epochs is a rather long time. Could the authors convince me this is a standard setting by providing some reference?


================ after rebuttal=========================
Thank the authors for reply. My rating does not change. The proposed does look similar to XNOR, and the only difference seems to be how the scales are updated. Since there is only one scale per layer, I will be quite surprised if the proposed method can be much better than XNOR. Moreover, since BatchNorm is not quantized and it is everywhere in a ResNet-like architecture, it surprises me how much the scale helps. Finally, I am worried about practical benefits towards the authors' claim because the networks are not fully quantized.


**Experience Assessment:**

I have published one or two papers in this area.

**Review Assessment: Checking Correctness Of Derivations And Theory:**

N/A

**Review Assessment: Checking Correctness Of Experiments:**

I assessed the sensibility of the experiments.

**Review Assessment: Thoroughness In Paper Reading:**

I read the paper at least twice and used my best judgement in assessing the paper.

---

> ### Author Response · Authors · 2019-11-13
> **Response to review**
>
>
> We have carefully read through the manuscript to look for and fix grammatical issues.  If the reviewer points out any specific issues that remain, we will be happy to fix them to improve the manuscript.
>
> There is one step size for each layer of weights (that is, for each weight tensor) and one step size for each layer of activations.  This is noted in section 2.1 with the line "each layer of weights and each layer of activations has a distinct step size".  BatchNorm is handled using full precision operations.  We have updated the text in section 2.3 to clarify this point.
>
> The proposed method offers 2 main benefits.  First, it allows for a reduction in model size, facilitating the deployment deep networks operating on the edge (for example, in mobile phones).  Second, it points to the value of developing next generation inference hardware optimized for low precision operations to improve throughput and energy efficiency, while reducing latency.  As we also note in our response to Review #2, we believe it is the correct first step to demonstrate the ability of deep networks to achieve high accuracy at these extremely low precisions before inference chips optimized for these precisions is commercially available.  As such hardware becomes available, we look forward to benchmarking these low precision networks against full precision alternatives.
>
> We now highlight the importance of XNOR in the introduction by noting it provided the first demonstration of tuned quantization.  XNOR differs from our approach in that the former employs a quantizer tuned based on data statistics, in approach later taken by techniques such as TWN, Dorefa, FAQ, and Half-wave gaussian quantization, while the latter learns the quantization through backpropagation, which is noted in the introduction.
>
> Providing a comparison between various quantization methods using exactly the same weight decay or learning rate schedule is difficult, as there is no standardization across prior papers on settings.  Properly re-implementing and re-running methods from prior papers can be challenging if even trivial details (for example, initial values) are missing from the original publications, and even if done properly, it is possible that different weight decay values or schedulers are optimal for different methods.  However, we have sought to address this as best we can by showing our results at different weight decay values and showing the simple power of 2 search that was employed to find the weight decays chosen (Table 2).  Notably, our approach still out performs the next best method even when using a weight decay of 10^-4, which is fairly standard for a full precision ResNet-18.  We also ran an experiment using a more traditional step-based weight decay (Section 3.5), and found that again our approach still out performs that next best method (with only 90 epochs of fine-tuning, compared to a total of 360 epochs of fine-tuning for the next best method).
>
> If of interest, Table 3 explores the value of the gradient adjustment derived from Equation 4, and shows that it is important for achieving high accuracy.
>
> Relatively long fine-tuning in the quantized space after initializing from a full precision trained network is fairly common when targeting networks with less than 8-bits of precision, and in fact the 90 epochs we employ is on the short side.  For example  Choi et al. ("Learning low precision deep neural networks through regularization") fine-tuned for 100 epochs, McKinstry et al. ("Discovering low-precision networks close to full- precision networks for efficient embedded inference") fine-tuned for 110 epochs, Baskin et al. ("Nice: Noise injection and clamping estimation for neural network quantization") fine-tuned for 120 epochs, and Jung et al., ("Joint training of low-precision neural network with quantization interval parameters") used a progressive quantization approach that amounted to a total of 360 additional epochs of fine-tuning for 2-bit models.

---

### Official Review · AnonReviewer2 · 2019-11-01
**Official Blind Review #2**

**Rating:** 6

**Review:**

The motivation of the paper is to be able to train low precision networks to a high-accuracy. Quantization is a useful tool in model compression, and doing it well for very low-precision models (2-3 bit precision specifically), is challenging.

The main contribution of the paper comes from:
a) Step Size Gradient: They propose a gradient which is sensitive to the distance between the value and the transition point. This is different from other methods which have gradients dependent only on the clip point.
b) Step Size Gradient Scale: This is an interesting contribution, where they try to match the ratio of average update of the step size ‘s’ and average magnitude of ‘s’, with that of the network weights. This leads them to scale the gradient according to the precision and number of parameters. They demonstrate that this scaling actually helps improve the accuracy.

The results for 8-bit precision are not new. Several results (Quantization and Training of Neural Networks for Efficient
Integer-Arithmetic-Only Inference, Jacob et al., Quantizing deep convolutional networks for
efficient inference: A whitepaper, Krishnamoorthi et al.), show 8-bit quantization results where the accuracy matches floating point accuracy, and in some case exceeds it (low precision quantization acting as a regularizer). However, the results for lower precision are impressive.

There are a few questions:
1. In sec 2.1, you mention that ‘each layer of weights and activations has a distinct step size, represented as an fp32 value, initialized to …’. Can you explain the intuition behind the initial value of the step size, and how is it a function of v?
2. ‘Model Compression via Distillation and Quantization’ (Polino et al.) shows distillation actually helps significantly improve accuracy. I wonder if the authors have tried different weight combinations for the distillation loss, and using bigger models as teacher models.
3. I would like to get more details of the inference setup, specifically the size and inference latency improvements over full-precision networks. The practical applicability of low-precision networks, specifically 2-bit and 3-bit networks, equally depends on the inference infrastructure, as it does on the training improvements.
4. Have you evaluated your method for a non-Vision usecase?

Overall this is a good work, I would tend towards accepting this.


**Experience Assessment:**

I have read many papers in this area.

**Review Assessment: Checking Correctness Of Derivations And Theory:**

I did not assess the derivations or theory.

**Review Assessment: Checking Correctness Of Experiments:**

I assessed the sensibility of the experiments.

**Review Assessment: Thoroughness In Paper Reading:**

I read the paper at least twice and used my best judgement in assessing the paper.

---

> ### Author Response · Authors · 2019-11-13
> **Response to review**
>
>
> Question 1:
> We expect that a step size that provides a reasonable quantization of some data, "v", should scale with the magnitude of that data (captured by "<|v|>"), so that the values are reasonably spread across the bins of the quantizer.  We also expect that as the number of quantized states increases, the step size itself should decrease, so that the data can be quantized more finely (captured by "1/sqrt(Q)").  The particular heuristic we chose worked well in practice to give a reasonably good initial quantization that could then be further improved through training.
>
>
> Question 2:
> Knowledge distillation for quantized networks is certainly an interesting area for research, but we didn't want to expand the focus of our manuscript too far beyond the quantizer training method proposed, particularly as Polino et al. and Mishra and Marr already have provided a good overview of knowledge distillation in this domain.  Thus, we chose to show only the simplest knowledge distillation setup we are are aware of, distilling from a full precision teacher to a low precision student network with the same architecture.  This approach offers has the benefit that the same trained high precision network used for weight initialization also serves as our teacher, and thus no additional networks are required, whereas distilling from a larger architecture would require training (or otherwise having access to) an additional network.  However, we have since run an experiment to look at different weights on the distillation loss, and now note in section 3.7: "we used the distillation loss function of Hinton with temperature of 1 and equal weight given to the standard loss and the distillation loss (we found this gave comparable results to weighting the the distillation loss two times more or less than the standard loss on 2-bit ResNet-18)."
>
>
> Question 3:
> To show the reduction in model size offered by lower precision models, we have modified Figure 3 to include full precision model sizes.
>
> We did not measure latency or related performance metrics on actual hardware, as our approach was developed in advance of commercially available inference hardware optimized for low precision operations.  Since algorithms are much cheaper to develop than new hardware, we believe it is the correct first step to demonstrate the ability of deep networks to achieve high accuracy at these extremely low precisions before hardware is created to optimize for these precisions.  As new low precision inference chips become available, we look forward to benchmarking these low precision networks against full precision alternatives.
>
>
> Question 4:
> While we have not done so yet due to time constraints, we anticipate evaluating this approach for domains beyond vision as a next step in our research.

---

### Author Response · Authors · 2019-11-13
**Thank you to reviewers**

We thank all reviewers for their time and thoughtful comments.  We have done our best to address each point both in our comments to each reviewer and, where indicated, with improvements to the manuscript itself.

---

### Public Comment · ~Fabien_Cardinaux1 · 2019-12-21
**Related Work on Learning Quantization Parameters**

Dear Authors,

Congratulations for having your paper accepted. I enjoyed very much reading it. It is nicely written and the results are impressive.

I would like to attract your attention to our related work which was also submitted and accepted to ICLR2020.
https://openreview.net/forum?id=Hyx0slrFvH
Basically, we show that, with a careful parametrization of the quantizer, we can learn the step size and the bitwidth for each layer in order to achieve a mixed precision DNN for a given size budget.

I am looking forward to meet you and discuss deep neural nets quantization with you in Addis Ababa.

---

> ### Author Response · Authors · 2020-01-03
> **Interesting paper**
>
> Thanks for bringing this to our notice.  Your paper is interesting work, and it's nice to see quantizer bit width tuning getting some attention.  Looking forward to further discussion!

---

### Decision · Program_Chairs · 2019-12-19

**Decision:**

Accept (Poster)

**Comment:**

Main content: Paper is about training low precision networks to a high-accuracy.

Discussion:
reviewer 2: impressive results, main questions are around some clarity in the experiments tried, but sounds like authors addressed most of this in rebuttal.
reviewer 1: well written paper, but authors think some technical details could be clarified.
reviewer 3:  well written but experimental section could be improved.
Recommendation: all reviewers are in consensus, well written paper but some experiments/technical details could be improved. i vote poster.